# EFFICIENT QUANTUM STATE RECONSTRUCTION USING UNSUPERVISED LEARNING FOR QUANTUM CIRCUIT CUTTING

## ABSTRACT

Current quantum computer (QC) fabrication encounters challenges when attempting to scale up the number of qubits. These challenges include errors, physical limitations, interference, and various other factors. As a remedy, quantum circuit cutting holds the promise for studying large quantum systems with the limited qubit capacity of quantum computers today. With quantum circuit cutting, the output of a large quantum circuit could be obtained through classical post-processing of fragmented circuit outputs acquired through different measurement and preparation bases. However, such reconstruction process results in exponential quantum measurement cost with the increase in the number of circuit cuts. In this paper, we demonstrate efficient state reconstruction using a Restricted Boltzmann Machine (RBM) with polynomial resource scaling. We explore the benefits of unsupervised learning for simulating extensive quantum systems, exemplified by the reconstruction of highly entangled multi-qubit Greenberger–Horne–Zeilinger (GHZ) states from fragmented circuits as well as output states from random unitary circuits. Our experiments illustrate that fragmented GHZ circuits, at the state-of-the-art scale of up to 18 qubits, can be reconstructed with near-perfect fidelity using only 100 sample measurements compared to $4^{18}$ sample measurements needed otherwise.

## 1 INTRODUCTION

Large-scale quantum simulations provide a boost to the development of not only next-generation advanced technologies but also in advancing fundamental science Grover (1996); Shor (1997); Robert et al. (2021). For instance, large-scale quantum simulations have been used in various applications, from novel drug discoveries in healthcare to improving risk-modelling in finance. However, a key challenge in bringing these applications to market at present is the lack of required simulation resources and the unavailability of large-scale quantum computers (QC). QCs with a large number of qubits could solve practical problems but require significant time and effort to build. Furthermore, the low error tolerance of these systems limits the quality of solutions attainable.

Despite their limited resources, the state-of-the-art *small-scale* quantum computers offer some respite to these promising applications. QCs with a lower number of qubits are already easily available through popular cloud platforms and have acceptable error tolerance on their outputs. In order to benefit from these properties, large quantum circuits can be implemented on small-scale QCs using a method called *quantum circuit cutting*, wherein a large circuit is *cut* into fragments of smaller circuits Bravyi et al. (2022). Enhanced scaling of quantum computing can be efficiently achieved using these smaller fragments that run independently on existing quantum computers Peng et al. (2020). Fragmentation also mitigates many hardware issues related to gate imperfections, read-out errors and decoherence. Ayral et al. (2020; 2021).

In recent literature, classical post-processing has been adopted to reconstruct the output of the full quantum circuit from the results of the fragmented circuits Lowe et al. (2023); Chen et al. (2022); Uchehara et al. (2022). However, the reconstruction of multi-qubit states from the underlying circuit, also called *quantum state tomography* (QST) is a challenging and compute-intensive problem. Both (1) the classical processing time, and (2) the number of quantum measurements required to re-

combine fragments scale exponentially with the number of cuts; the poor scalability severely limits the practical usefulness of quantum circuit cutting.

In order to mitigate classical post-processing overhead, different methods have been proposed in recent years. Stochastic methods through randomized measurements Lowe et al. (2023), sampling Chen et al. (2022) along with optimizing Uchehara et al. (2022), and tailoring cut points Chen et al. (2023a) were shown to reduce the time complexity. Furthermore, methods with maximum likelihood fragment tomography (MLFT) have demonstrated the reduction of the computation overhead from exponential to sub-exponential with finite-shot error mitigation Perlin et al. (2021). Chen *et.al.* used classical shadow tomography with fewer measurements to predict the outcome of the circuit Chen et al. (2023b).

QST also requires a significantly large number of quantum measurements: *e.g.* to accurately reconstruct an 8-qubit quantum state, 1 million measured samples are required. The sample complexity can be reduced by different stratergies Haah et al. (2017); Qin et al. (2023). As QST is a data-driven problem, machine learning (ML) methods have been employed to tackle the resource complexity. In particular, using the variational ansatz of neural network quantum states and Restricted Boltzmann Machines (RBM), efficient reconstruction of quantum states for up to $N = 80$ qubits was successfully demonstrated from the measurement data of the full circuit Torlai et al. (2018). Similarly, other neural network topologies such as CNN Schmale et al. (2022); Lohani et al. (2020), RNN Quek et al. (2021) and GA Ahmed et al. (2021); Zhong et al. (2022); Cha et al. (2021); Zhu et al. (2022) along with RBM Carrasquilla et al. (2019) have been explored to provide a feasible reconstruction of quantum states. However, ML methods until now have only been applied to reconstruct the full quantum state, *i.e.,* the output of a full quantum circuit. We extend the use of machine learning-assisted tomography to reconstruct fragmented circuit states with manageable classical resources. Unlike classical circuits, quantum circuits generate entanglement among the qubits, making the reconstruction of states post circuit-cutting a complex task.

The objective of our paper is to apply unsupervised learning to fragmented circuits and reconstruct the *full state* of a larger quantum circuit in an efficient and scalable manner. In particular, we train RBM to efficiently reconstruct the probability amplitudes of the output from circuit fragments. The full state of the circuit is then constructed by applying a tensor product formalism on the output states of all the fragments. Using the prototype of a highly-entangled, multi-qubit Greenberger–Horne–Zeilinger (GHZ) circuit, we show that the corresponding highly entangled GHZ states can be constructed with high fidelity using very few measured samples of fragmented circuits. To generalize our results, we also test our method on random unitary circuits. Our paper has the following contributions:

- We propose efficient machine learning assisted fragment tomography to reconstruct the quantum states from fragmented quantum circuits.

- Our proposed method can significantly reduce the fidelity loss due to the increase in complexity with number of qubits / cuts with scalable classical resources.

- Experiments are conducted demonstrating the effectiveness of our approach by successfully reconstructing highly entangled GHZ circuits with up to 18 qubits using only 100 sample measurements, achieving near-perfect fidelity.

## 2 RELATED WORKS

The closest work related to our paper uses the Maximum-Likelihood Fragment Tomography (MLFT) as an improved circuit cutting technique to mitigate the hardware noises Perlin et al. (2021). They used clustered random unitary circuits to demonstrate the reconstruction of up to 18 qubits with 0.99 fidelity. However, their reconstruction procedure requires 1 million samples and a sub-exponential scaling of the number of samples was observed with the increase in circuit size.

In Chen et al. (2022), efficient state reconstruction of subcircuits up to size 10 qubits was shown using Monte-carlo sampling. Similarly, subcircuit reconstruction of 5 qubits was shown by using optimal cut points in Chen et al. (2023a). Also, the expectation values of observables were studied by cutting up to 8 qubit circuits in Uchehara et al. (2022); Chen et al. (2023b).

Another important work is using randomized measurements Lowe et al. (2023) by which Quantum Approximate Optimization Algorithm (QAOA) circuits of up to 13 qubits were executed by circuit cutting. Also, variational energy of 129 qubits was estimated by circuit cutting. However, we note that the problem of state reconstruction was unaddressed and only expectation values were estimated. Also, exponential scaling of resources was required to obtain the results.

## 3 THEORY

### 3.1 BACKGROUND OF QUANTUM CIRCUITS

In classical computing, the information is stored in the form of binary digits called bits. The quantum counterpart of a classical bit is a qubit (short form of quantum bit) that takes on the two basis states $|0\rangle$ and $|1\rangle$. Unlike the classical bit, a qubit can simultaneously be in a superposition state represented as a linear combination of probabilities in corresponding basis states, $i.e.,$ $|\psi\rangle = \alpha|0\rangle + \beta|1\rangle$ with amplitudes $(\alpha, \beta) \in \mathbb{C}$. Here, the probability conservation requires $|\alpha|^2 + |\beta|^2 = 1$. With $n$ qubits, a superposition of $2^n$ binary combinations is possible, each with a specific amplitude.

Quantum programs are essentially quantum circuits expressed as a collection of quantum gates acting on qubits. Execution of the circuit involves applying quantum gates on qubits. The quantum gates used in our paper are the Hadamard (H) gate and the CNOT gate as shown in Fig. 1, 2 and 4. H-gate is a single qubit gate that transforms the qubit state to the superposition of base states. CNOT-gate acts on two qubits and flips the target qubit (indicated by symbol $\oplus$) if the controlled qubit (indicated by symbol $\circ$) is in state $|1\rangle$; $|0\rangle$ state of the controlled qubit leaves the target qubit unchanged. When a gate acts on two or more qubits, the qubits become entangled: their states can only be represented collectively rather than individually. Consequently, a system of $n$ entangled qubits encodes $2^n$ output states simultaneously, giving quantum computers immense representation capacity compared to the equivalent classical computers which require $2^n$ bits. For example, in the 3-qubit system of Figure 1 (left), all three qubits are entangled by the controlled-NOT gates that correlate the states of each qubit together. Hence, there are $2^3 = 8$ potential output states ($|000\rangle, |001\rangle, ... |111\rangle$) that can be observed when the output qubits are measured.

A quantum gate performs a trace-preserving linear operation on *all* quantum states of the system. When a gate is applied to any one of the $n$ entangled qubits, the probabilities of all $2^n$ states are transformed. The effects of entanglement make the division of qubits into independent sub-circuits non-trivial, as inter-circuit correlations cannot be maintained without proper circuit-cutting methods.

### 3.2 BASICS OF QUANTUM CIRCUIT CUTTING

Larger quantum circuits can be cut into smaller sub-circuits that can be independently executed on quantum computers with limited qubits. There are different ways of cutting the quantum circuit such as wire cutting and gate cutting. In this paper, we focus on the quantum wire cutting where the wire carrying the quantum information of a qubit is divided into sets of measurement and state-preparation operations.

Consider a single qubit - The state of the qubit can be completely characterized by the density matrix $\rho \in \mathbb{C}^{2 \times 2}$ representing qubit state probabilities:

$$\rho = \frac{1}{2} \sum_{i=1,8} c_i Tr(\rho O_i) \rho_i, \tag{1}$$

where $O_i$'s (*e.g.* $O_1$ and $O_2$ on $I$-axis) are the Pauli operators $I, X, Y, Z$ of the different base-axis with their corresponding eigenprojections $\rho_i$ and their eigenvalues $c_i$. $Tr$ represents the trace operation and other variables follows as, $c_i = \pm 1$, $O_1 = O_2 = I$, $O_3 = O_4 = X$, $O_5 = O_6 = Y$, $O_7 = O_8 = Z$, $\rho_1 = \rho_7 = |0\rangle\langle 0|$, $\rho_2 = \rho_8 = |1\rangle\langle 1|$, $\rho_3 = |+\rangle\langle +|$, $\rho_4 = |-\rangle\langle -|$, $\rho_5 = |+i\rangle\langle +i|$ and $\rho_6 = |-i\rangle\langle -i|$.

Each term $Tr(\rho O_i)\rho_i$ in the Eq. 1 can be divided into two parts. The first part, $Tr(\rho O_i)$ is the measurement outcome *i.e.,* the expectation of the observable $O_i$ when the qubit is in the state $\rho_i$. The second part is the initialization of the qubit or the preparation of its eigenstate $\rho_i$. Wire cutting of quantum circuit is based on Eq. 1.

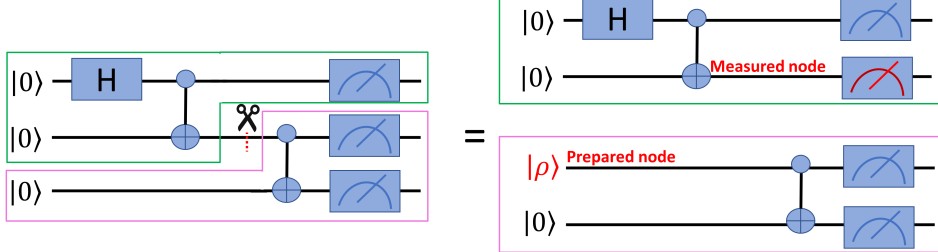

Figure 1: Fragmentation of a three-qubit GHZ circuit into two parts. Left part shows the full circuit and the point of the cut whereas right part shows the two fragmented subcircuits. The first fragment outlined with green color constitutes the first and second qubit while the second fragment highlighted in the pink box constitutes the second and third qubit.

Consider a $n$-qubit quantum circuit fragmented into two halves. The state of the full circuit $\rho \in \mathbb{C}^{2^n \times 2^n}$ can be written as the tensor product of the states of two fragments as

$$\rho \simeq \frac{1}{2} \sum_{i,j=1,8} c_i c_j \rho_1(O_i) \otimes \rho_2(O_j). \tag{2}$$

The fragmented state in Eq. 1 consists of two terms: $\rho_1(O_i)$ and $\rho_2(O_j)$ representing the measurement and state preparation terms, respectively. From Eq. 2 it follows that each $\rho_2(O_j)$ is a conditional state obtained after the measurement of the first fragment to state $\rho_j$ or preparing qubit in the state $\rho_j$. An illustration of the fragmentation of a three-qubit GHZ circuit in Fig. 1. Here, the circuit is cut at the second qubit. The first fragment is shown in the green box while the second fragmented circuit is denoted by the pink box. It is clear from the figure that fragmentation always introduces an extra qubit. However, we note that fragmented circuits are smaller in general. Additionally, they can be executed independently. In effect, fragmentation enables complexity savings by reducing the circuit size.

As shown in the figure, for the first fragment the cut appears at the end of the circuit and hence the second qubit is measured in different bases $X, Y, Z, I$. Similarly, the cut appears at the front of the second fragment and hence the corresponding qubit is prepared in a variety of initial states $\rho_j$. In short, each fragment has several different possibilities with each of them either prepared in some initial states and measurement bases. It is clear from Eq. 2 that for a circuit cut into two halves, there are a total of $16(4^2)$ possibilities. In general, classical post-processing requires a cost of $4^K$ for a circuit with $K$ cuts.

### 3.3 FRAGMENT TOMOGRAPHY

One alternative to reducing the complexity is to use the fragments as objects rather than their conditional states $\rho_{1,2}$ and perform the fragment tomography on these objects Perlin et al. (2021). A fragmented circuit of $n$-qubits will have $n_i$ quantum inputs and $n_o$ quantum outputs at the cut place in addition to the usual inputs and the *classical* outputs. By treating the quantum circuit as a channel, we can rewrite the action of the circuit as a four-partite state as

$$\Lambda \equiv \sum_{k,l;m,np,q;r,s} |k\rangle\langle l| \otimes |m\rangle\langle n| \otimes |p\rangle\langle q| \otimes |r\rangle\langle s|, \tag{3}$$

where $p, q$ ($r, s; k, l; m, n$) index states in classical input (classical output, quantum input, quantum output) respectively. Using the initial state as $|00...0\rangle$ as well as the computational state for the classical output we can simplify the channel state as

$$\tilde{\Lambda} \equiv \sum_{k,l;p,q;s} |k\rangle\langle l \otimes |p\rangle\langle q| \otimes |s\rangle\langle s| \tag{4}$$

$$= \sum_s \tilde{\Lambda}_s |s\rangle\langle s|$$

To characterize the output of the fragmented circuit, we need to approximate the probability $\tilde{\Lambda}_s$ of obtaining the bit string $s$ on the classical output from all the experiments with a variety of quantum inputs to the circuit and measuring the quantum outputs in different bases. In our work, we use unsupervised learning with RBM to approximate this probability distribution and reconstruct the states $\rho_{1,2}$ of the fragmented circuits.

## 4 METHOD

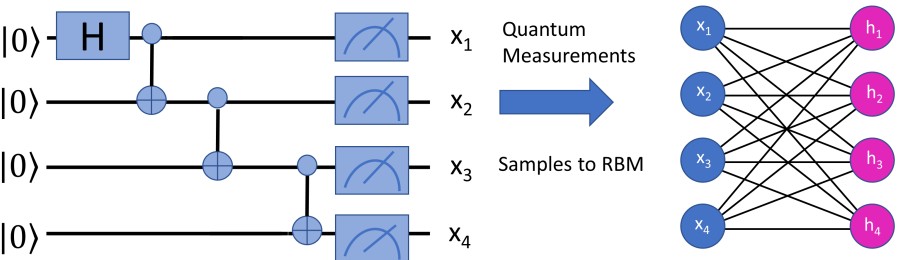

Figure 2: RBM parameterization of the quantum state $\psi$. Measurements from a quantum circuit of $N = 4$ qubits $\{x_1, x_2, x_3, x_4\}$ are fed as samples to RBM of $N = 4$ visible nodes. Hidden layer $h_i$'s acts as auxiliary qubits. After training, the distribution of RBM as given in Eq. 6 approximates the output state of the quantum circuit. Note that for the computational basis ($Z$ or $I$ basis) $x_i$ and $h_i$ take values of 0 and 1.

We use the neural network (NN) architecture of RBM as shown in Fig. 2 in our study. RBM features two binary stochastic layers - one visible layer $\mathbf{x} = \{x_i\}$ consisting of the physical qubits and one hidden layer $\mathbf{h} = \{h_i\}$ of neurons that are fully connected to the visible layer with some weighted edges $W_{ij}$. The expressive power of the NN is determined by the ratio $\alpha = M/N$ between the number of hidden $M$ to visible neurons $N$. The associated probability distribution of RBM is given by the Boltzmann distribution as

$$p(\mathbf{x}, \mathbf{h}) = \frac{1}{Z} e^{\sum_j b_j x_j + \sum_i c_i h_i + \sum_{i,j} W_{ij} x_j}, \tag{5}$$

Here, $b_i$ and $c_i$ are the biases of visible and hidden layers respectively, $W_{ij}$ are the weights and $Z = \sum_x p(\mathbf{x},\mathbf{h})$ is the normalization constant. The distribution over the visible nodes can be obtained by the marginalization of hidden degrees of freedom as

$$p(\mathbf{x}) = e^{\sum_j b_j x_j + \sum_i log(1 + c_i + \sum_j W_{ij} x_j)}. \tag{6}$$

### 4.1 QST USING RBM

A quantum state of $n$-qubits in some reference basis $\mathbf{x}$ (with $|x\rangle = |x_1 x_2..x_n\rangle$) can be represented by RBM as $\psi_{RBM}(\mathbf{x}) = \sqrt{p(\mathbf{x})}/Z$ with the probability distribution given in Eq. 6. We chose the computational basis for which $x_i, h_i = [0, 1]$.

Measurements in a variety of bases $b = \{X, Y, Z\}$ will be distributed according to the probabilities $p^{b(x)} \propto |\psi^b(x)|^2$. RBM of some internal parameter $\kappa$ can be tuned in to approximate the wavefunction to the probability distribution in each of the bases $\psi_k^b(x) \equiv |\psi^b(x)|^2$. This can be done by minimizing the total statistical divergence $\Xi(\kappa)$ between the targeted and the reconstructed distributions. In particular, we use Kullbach-Liebler (KL) divergence defined as

$$\Xi(\kappa) \equiv \sum_b \mathbb{KL}(\kappa) = \sum_b \sum_{x^b} p(x^b) log \frac{p(x^b)}{|\psi(x^b)|^2}. \tag{7}$$

The total divergence $\Xi(\kappa)$ is positive definite and attains a value of 0 when the reconstruction is perfect.

## 4.2 FRAGMENTED SAMPLING

Our method of reconstructing the states of the full circuit is depicted in Algorithm 1. Our study uses a multi-qubit GHZ circuit as the model. First, we cut the ciruit into $K$ number of fragments which can be executed independently. For each fragment, measurement samples are collected by running the fragmented circuits on a quantum simulator called "QUIMB" Gray (2018). We fix the computational basis as the reference basis and $|000...0\rangle$ as the initial state in our simulations. Specifically, for a circuit fragmented into two halves (as given in Fig. 1), we collect samples with measurement bases $\{III...I, III..X, III...Y\}$ for the first fragment. The samples for the second fragment are collected by inputting different states $\{|000...0\rangle, |100...0\rangle, |+00...0\rangle, |-00...0\rangle, |+i00...0\rangle, |-i00...0\rangle\}$ and measuring the circuit in the corresponding bases $\{III...I, XII..I, YII...I\}$. We initialize the biases, weights and learning rate of RBM. The collected samples from the simulator are fed to the RBM. RBM is then trained to learn the action of the circuit as given in Eq. 4. Training of RBM is done for a fixed number of epochs $N_e$ by minimizing the KL divergence such that the probability distribution of the output of the fragmented circuit is obtained. In particular for each epoch, we calculate $KL$ divergence given in Eq. 7 and gradients of the biases and weights (Eq. 10) using stochastic gradient descent method. The updated parameters are then used to calculate the RBM wavefunction. Once, the training is completed for $N_e$ epochs, we sample through the $|\psi_{RBM}|^2$ using Gibbs sampling and construct probability amplitudes of fragmented circuit. Once an accurate distribution of the sub-circuit is obtained, the full state can be determined from the tensor product of the fragmented probability distribution.

---

**Algorithm 1:** Fragmented circuit state reconstruction

**Input:** $N_s$=number of samples, $K$ = number of cuts, $N_e$=number of epochs
**Output:** $\psi$ Amplitudes of reconstructed state

1 **while** $K \neq 0$ **do**
2     Get $N_s$ measurement data using QUIMB simulator;
3     Initialize RBM (learning rate, weights, biases);
4     **for** $j = 1$ **to** $N_e$ **do**           // Training of fragmented circuits
5         Calculate KL divergence;
6         Update gradients;
7         Update RBM wavefunction $\psi_{RBM}$;
8     Sample through $|\psi_{RBM}|^2$ using Gibbs sampling and construct probability amplitudes of fragmented circuit;
9 Reconstruct full circuit state amplitudes $\psi$ using tensor product formalism

---

## 5 RESULTS

We now illustrate the efficiency of our method using a GHZ circuit generator. GHZ states are a highly entangled states that are central for quantum metrology Omran et al. (2019) and quantum error correction Nielsen & Chuang (2010). A multi-qubit GHZ state $\psi_{GHZ}$ is an equal superposition of all qubits in the up and down state as

$$|\psi_{GHZ}\rangle = \frac{1}{2}(|000...0\rangle + |111...1\rangle).$$ (8)

The efficiency of our method in reconstructing the states is quantified by calculating the overlap between the GHZ state $\psi_{GHZ}$ and the RBM wavefunction $\psi_{RBM}$ defined in Eq. 6. For this, we use the measure fidelity $\mathcal{F}$ defined as

$$\mathcal{F} = |\langle \psi_{RBM} | \psi_{GHZ} \rangle|^2.$$ (9)

When the reconstruction is perfect, fidelity $\mathcal{F} = 1$.

*Experimental setup*: We use an RBM of $N$ visible nodes to train fragmented circuit of $N$ qubits. Number of hidden nodes are also chosen to be equal to $N$. Zero bias is used for the hidden layer whereas the bias $c_i$ of the visible layer is chosen from uniform random distribution in the interval $[0, 1]$. Similarly, the values of weights $W_i j$ are chosen randomly from the uniform distribution in

the interval $[-0.1, 0.1] \times \sqrt{6/2N}$. Stochastic gradient descent is used to update the neural network parameters $\kappa(l) = (c_i(l), W_{ij}(l))$ with KL divergence defined in Eq. 7 as the cost function. The updated parameters at the $l$th iteration are given by

$$\kappa(l+1) = \kappa(l) - \gamma(l)D(l), \tag{10}$$

where $D(l)$ is the gradient averaged over the samples (for more details, see Torlai et al. (2018)) and $\gamma(l)$ is the learning rate fixed to 0.5 for the first half of the training after which it is decreased to 0.05.

### 5.1 SINGLE CUT

First, we consider a single cut such that the circuit is divided into two equal halves as shown in Fig. 1. The two fragmented circuits are then trained independently using unsupervised machine learning with RBM.

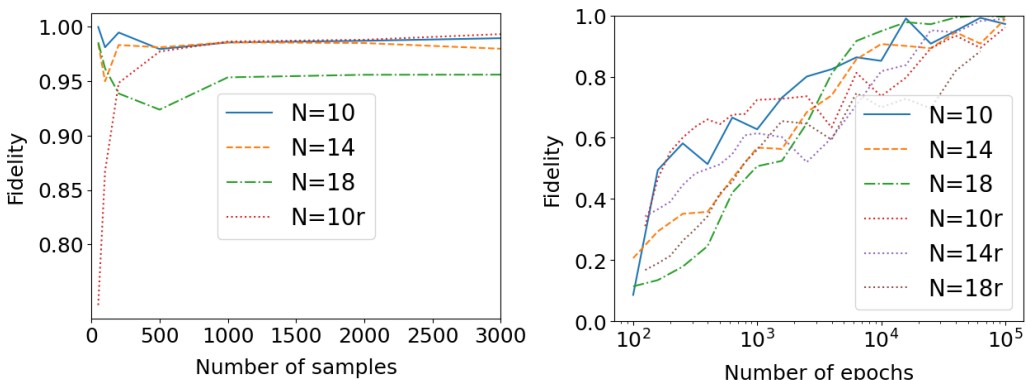

Figure 3: Fidelity as a function of sample size (left panel) and number of epochs (right panel) for GHZ circuit of $N$ qubits fragmented into two equal halves. Here, blue solid lines are for $N = 10$ qubits, brown dashed lines for $N = 14$ and green dot-dashed lines for $N = 18$ qubits for GHZ circuit and red, purple and brown dotted lines are for $N = 10, 14, 18$ qubits for random unitary circuit. For the left panel, the number of epochs is fixed to 10000 whereas for the right panel, 100 samples are used for GHZ circuit and 1000 samples for random unitary circuit. The figure demonstrates the efficiency of using neural network training for state reconstruction with scalable resources.

From Eq. 1, it is clear that at least four measurements are required to determine the state of a single qubit. In general, for an $N$ qubit system one needs to determine $4^N$ unknown parameters which require at least the same number of measurements. For instance, a 10 qubit system requires $10^7$ (10 million samples). However, our fragmentation scheme can reconstruct the state with 0.99 fidelity using only 100 samples. This is illustrated in Fig. 3.

The left panel of Fig 3 shows the dependence of fidelity on sample size. As the plots are for single instance of experiment, a drop in fidelity is seen with sample size for GHZ circuit. We note that 2% variance in the results are obtained while repeating the experiments. Here, the number of training steps (epochs) is fixed at 10000. When the number of qubits in the circuit is increased, the fidelity also drops. This can be attributed to the increase in the complexity of the circuit as the number of qubits increases. However, we observe that with a small sample size of 100, one can get very near to perfect fidelity by increasing the training steps. This is clear from the right panel of Fig. 3 where the plot of fidelity vs. number of epochs is shown for a sample size of 100. Our result also demonstrates the power of RBM as a good sampler. Indeed, we see that even after almost doubling the size of the circuit from 10 to 18 qubits, the number of samples required to accurately reconstruct the data remains the same. Note that an exponential scaling of resources is required to reconstruct the state exactly. The generalization of our method is demonstrated by effeciently reconstructing the state using random unitary circuit in Fig. 3.

## 5.2 Multiple cuts

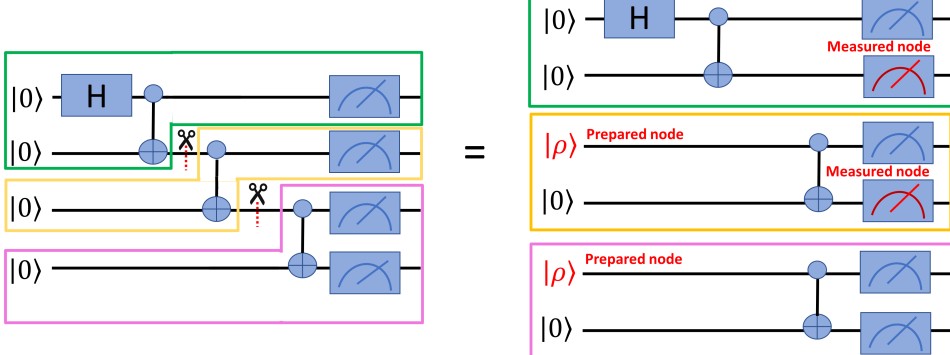

Figure 4: An example of cutting 4 qubit GHZ circuit into three parts. The full circuit is shown on the left side whereas three fragmented sub-circuits are shown on the right. Here, the first cut is between the second and third qubit and the second cut is between the third and fourth qubit. The front fragment consists of the first and second qubit (green box), the middle fragment consists of the second and third qubit (yellow box) and the end fragment contains the third and fourth qubit (pink box).

Next, we turn to multiple cuts in the circuit. In general, any circuit with more than two cuts has three different kinds of structure: one front, middle, and end. This is illustrated for a 4 qubit GHZ circuit in Fig. 4. Here, the green boxed part consisting of the first two qubits is the front fragment, the middle fragment in yellow contains the second and third qubit and the pink color end fragments contains the third and fourth qubit. The front and the end structures are similar to the circuit with two cuts. Here, the last qubit of the fragmented circuit is measured for the front structure with the measurements done in $\{I...X, I...Y, I...I\}$ bases. For the end fragment, the first qubit is prepared in different initial states and measured in corresponding bases. For the middle fragment, the cut is at both the front and the end of the circuit. Hence, one needs to prepare the first qubit of the fragment in different initial states. In addition to this, the last qubit for the middle fragmented circuit has to be measured in different bases. In short, one needs sampling in $\{I...X, X...X, Y...X, I...Y, X...Y, Y...Y, I...I, X...I, Y...I\}$ bases for the middle fragment.

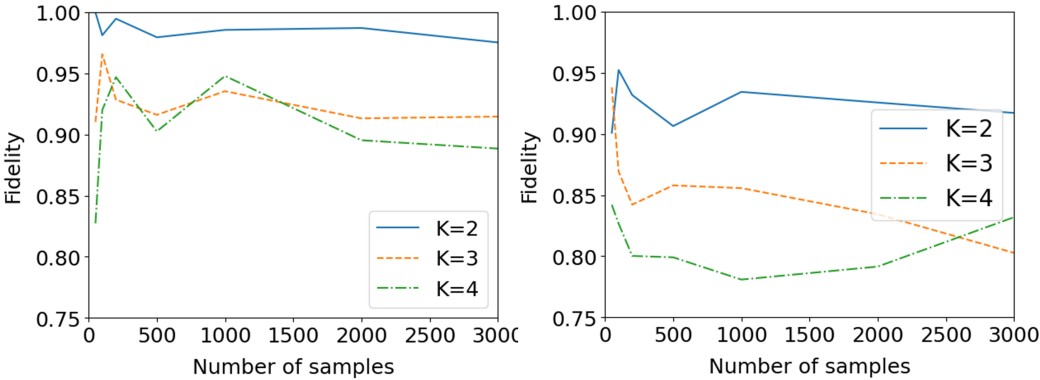

Figure 5: Comparison of fidelity with fragmentation. Here, the GHZ circuit is fragmented into $K$ cuts. The horizontal axis is for sample size and the vertical axis is for fidelity. The left panel displays the results for $N = 10$ qubit circuit and the right panel for $N = 18$ qubit circuit. Here, blue solid lines are for $K = 2$, brown dashed lines for $K = 3$ and green dot-dashed lines for $K = 4$ cuts. Fidelity drop with qubits and cuts illustrates the increase in complexity of the state reconstruction with increase in the number of qubits/cuts.

Figure 5 shows the reconstructed state's fidelity for multiple cuts. Here, $K$ denotes the number of pre-defined cuts. We study the circuits with $K = 2, 3, 4$. The left panel is for the GHZ circuit with $N = 10$ qubits and the right panel is for $N = 18$ qubits. For $N = 10$ qubit circuit, the cut is made at 5th qubit for $K = 2$. Circuit is cut at 4th and 7th qubit for $K = 3$ cuts and 4th, 6th and 8th for $K = 4$. Similarly, $N = 18$ qubit circuit is fragmented at 9th qubit for $K = 2$. For $K = 3$, the cuts are at 7th and 13th qubit. For $K = 4$, the circuit is sliced at 6th, 11th and 15th qubit. Here, the number of epochs is fixed at 10000. In both panels, we see that the fidelity drops significantly with the increase in the number of cuts. This is because of the exponential increase in complexity with the number of cuts. Also, the fidelity drop is larger in deep circuits (with more number of qubits) on account of growing complexity. For instance, the fidelity is around $0.9$ with $K = 4$ cuts for 10 qubit circuit (left panel) while it is only around $0.8$ for $N = 18$ qubit circuit (right panel). However, we see that the fidelity can be improved by increasing the training steps. This is illustrated in Fig. 6 where fidelity as a function of the number of epochs for training each sub-circuit is plotted. It is clear from the figure that using a very small sample size of $100$, we could obtain near-perfect fidelity by increasing the training steps for larger qubits as well as multiple-cut fragmented circuits.

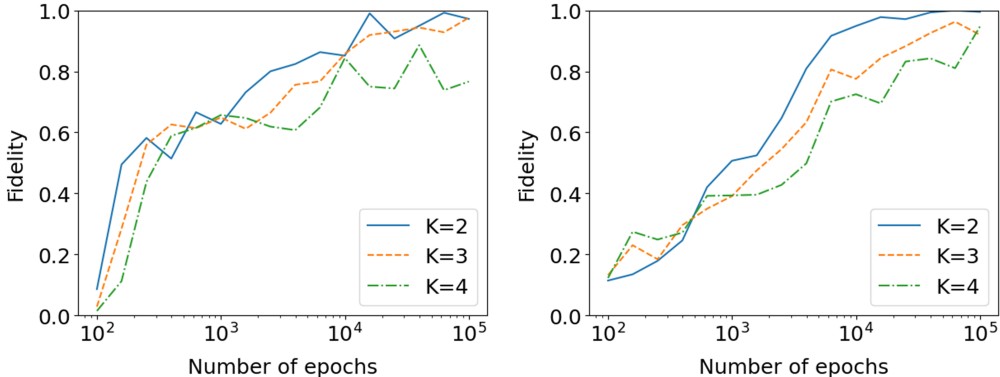

Figure 6: Fidelity versus number of epochs for $N = 10$ qubit (left panel) and $N = 18$ qubit (right panel) GHZ circuit fragmented into $K$ cuts. Here, blue solid lines are for $K = 2$, brown dashed lines for $K = 3$, and green dot-dashed lines for $K = 4$ cuts with sample size fixed to $100$. Results show that the fidelity can be significantly improved by increasing the training steps for fragmented tomography with large number of cuts.

## 6 CONCLUSION

Quantum circuit cutting is a promising method for large-scale, many-qubit quantum computation with currently available quantum computers. However, the state reconstruction procedure requires an exponential amount of classical resources, limiting the wide use of circuit cutting. In this work, we show how to overcome this issue using unsupervised learning with an RBM network architecture. We demonstrate that GHZ circuits of up to 18 qubits can be reconstructed with near-perfect fidelity using only 100 sample measurements. Though we observe a decrease in fidelity with an increasing number of qubits or cuts, we show that the fidelity loss can be greatly improved by increasing the number of training steps. We note that all calculations can be done using a laptop CPU in a few hours.

Another important feature of our proposed method is that requires only raw data of experimental snapshots of measurement rather than the estimation of the expectation value of observables. Also, the method estimates the wavefunction and overcomes the challenges of probabilistic approaches that demand positive definite distribution. Our method deals directly with wavefunction and can even be used to determine the phases of the state. We note that this is one of the unique features of our method compared to existing traditional approaches. In short, our study illustrates the importance of classical machine learning in quantum computing applications and reinforces the use of hybrid classical-quantum computing.

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
