# OpenReview forum: "EFFICIENT QUANTUM STATE RECONSTRUCTION USING UNSUPERVISED LEARNING FOR QUANTUM CIRCUIT CUTTING"
_ICLR.cc/2024/Conference — ICLR 2024 Conference Withdrawn Submission_

### Official Review · Reviewer_kuyP · 2023-10-29

**Soundness:** 2 fair
**Presentation:** 2 fair
**Contribution:** 1 poor
**Rating:** 3
**Confidence:** 4

**Summary:**

The paper tries to address the challenge of efficiently reconstructing quantum states from fragmented quantum circuits. Note that efficient state reconstruction is an essential subroutine in quantum information processing. The main method combines circuit cutting and Restricted Boltzmann Machines (RBM). The paper demonstrates that highly entangled GHZ states can be efficiently reconstructed with high fidelity using very few measured samples of fragmented circuits.

**Strengths:**

The paper explores a new possibility of quantum state reconstruction via circuit cutting. The example of the GHZ state looks interesting.

**Weaknesses:**

- There is no sufficient theoretical guarantee for the performance of the main method.
- Since the general quantum state has a much more complex structure than the GHZ state, it is not convincing that the method in this paper will have a good performance.
- The presentation is not good. Around Algorithm 1, it is difficult to find what a QUIMB simulator is. Also, it is hard to understand how to Update RBM wavefunction.
- The presentations and explanations for the numerical experiments are unclear and not convincing.

**Questions:**

- Why Eq. (1) only takes sum over i=1 and 8?
- What is the theoretical guarantee of this method for general pure and mixed entangled states?
- Why more samples do not lead to significantly better results in the numerical experiments?

**Details Of Ethics Concerns:**

NA.

---

> ### Author Response · Authors · 2023-11-21
>
> We thank the reviewer for his/her constructive comments. We have revised the paper to bring better clarity to our method and results. The main method of using RBM to represent quantum states efficiently follows from the reference Carleo_2017 which we have added. We also tested the circuit by inserting the random unitary gates distributed according to Haar measure.  We have modified the Fig.3 to add the results with random unitary gates and ensure that our method can accurately reconstruct even random states.
> Quimb is a quantum simulator that can be used to simulate quantum circuits to mimick the one run by quantum computers. We apologize for omitting the reference for quimb and has added in the revised version.
> The wavefunction is updated by updating the probability amplitude in Eq.(6) using Eq.(10). We have added reference Torlai 2018 that provides the details of updating wavefunction below Eq.(10).
> The density matrix of a qubit is a complex Hermitian matrix of size 2 by 2. Hence, it can be represented by Pauli matrices {I,X,Y,X}. Each of them have two eigenstates and this leads to a total of 8 parameters in the summation of Eq.(1).
> We note that estimating the theoretical limit is out of the scope of our paper.
> The fidelity was calculated only for one training instance. We note that the fidelity will vary over 3% over the average value when the experiment is repeated for GHZ states. We have modified the text in the paper to note this. For random unitary circuits, the fidelity increases with the increase in the sample size. This is because GHZ states has only two outcomes either all |00000..00> state or |111..111> state. However, with random unitary gates since the outcome can be random basis states like 00..0, 100..0, 010..0 etc. Hence, more samples helps to increase the fidelity. We have modified the Fig.5 to add the results with random unitary gates.

---

> > ### Comment · Reviewer_kuyP · 2023-11-22
> >
> > Thanks to the authors for the reply. However, I still have concerns about the theoretical guarantees of the methods, hence I will keep the evaluation unchanged.

---

> ### Author Response · Authors · 2023-11-22
>
> We thank the reviewer for his/her valuable response.  We note that RBMs are particularly well-suited for feature learning. The hidden layer captures features and patterns in the input data, and these features can be used for various tasks such as classification or generation. It showed impressive generalization abilities especially in complex and high-dimensional data, which can learn intricate patterns and representations that may be challenging for traditional methods. while RBMs may lack formal theoretical guarantees, its empirical success in various applications is undeniable [1] [2] [3] [4]. Our work is an empirical research, it is not reasonable for us to provide theoretical guarantee or proof.
>
> [1]Hinton, Geoffrey E., and Ruslan R. Salakhutdinov. "Reducing the dimensionality of data with neural networks." science313.5786 (2006): 504-507
>
> [2] Coates, Adam, Andrew Ng, and Honglak Lee. "An analysis of single-layer networks in unsupervised feature learning." Proceedings of the fourteenth international conference on artificial intelligence and statistics. JMLR Workshop and Conference Proceedings, 2011.
>
> [3] Carleo, Giuseppe, and Matthias Troyer. "Solving the quantum many-body problem with artificial neural networks." Science355.6325 (2017): 602-606.
>
> [4] Melko, Roger G., et al. "Restricted Boltzmann machines in quantum physics." Nature Physics 15.9 (2019): 887-892.

---

### Official Review · Reviewer_YKiP · 2023-10-29

**Soundness:** 3 good
**Presentation:** 3 good
**Contribution:** 1 poor
**Rating:** 3
**Confidence:** 4

**Summary:**

This manuscript introduces a RBM-based architecture designed to address the challenge of quantum state reconstruction. The authors present an unsupervised learning framework with different measurement results associated with fragmented circuits acquired by quantum circuit cutting. Furthermore, the authors show the model's effectiveness by employing it in the reconstruction of the 18-qubit GHZ state.

**Strengths:**

- This manuscript combines the methods of quantum circuit cutting and neural networks for quantum state reconstruction.

- The incorporation of RBM proves effective in reducing the required number of measurements for reconstructing the 18-qubit GHZ state compared with the previous approach that solely employed quantum circuit cutting.

**Weaknesses:**

- I believe the proposed model lacks novelty. Despite its combination of quantum circuit cutting and RBM, the whole framework still closely resembles the framework introduced in [1]. The authors don't cite this important previous work in their paper, let alone a comparison.

- The numerical experiment is exclusively focused on the reconstruction of the GHZ state. This singular focus gives the impression that the proposed method is primarily suited for GHZ state reconstruction. If this is indeed the case, it may restrict the model's applicability to other types of quantum states. However, if this is not the case, I suggest that the authors diversify their examples and provide a comprehensive discussion on the types of quantum states that can be effectively reconstructed using this proposed method.

  [1] Carrasquilla, J., Torlai, G., Melko, R. G., & Aolita, L. (2019). Reconstructing quantum states with generative models. *Nature Machine Intelligence*, *1*(3), 155-161.

**Questions:**

Major concerns:

- I have stated two major concerns in the "Weakness" section above.
- Another major concern is about the motivation of this work. In the Introduction, the authors assert, "We extend the use of machine learning-assisted tomography to reconstruct fragmented circuit states with manageable classical resources." It raises the question of why reconstructing fragmented circuit states is a critical thing, especially when many previous methods can accomplish state reconstruction without resorting to quantum circuit cutting. I suggest the authors to elaborate further on the motivation behind this work to provide a clearer context and rationale for this specific approach.

Minor comments:

- In the Introduction, the authors mentioned some previous work that leverage machine learning methods for quantum state learning.  However, they overlooked some noteworthy prior research, such as:

  Carrasquilla, J., Torlai, G., Melko, R. G., & Aolita, L. (2019). Reconstructing quantum states with generative models. *Nature Machine Intelligence*, *1*(3), 155-161.

  Zhu, Y., Wu, Y. D., Bai, G., Wang, D. S., Wang, Y., & Chiribella, G. (2022). Flexible learning of quantum states with generative query neural networks. *Nature Communications*, *13*(1), 6222.

  Cha, P., Ginsparg, P., Wu, F., Carrasquilla, J., McMahon, P. L., & Kim, E. A. (2021). Attention-based quantum tomography. *Machine Learning: Science and Technology*, *3*(1), 01LT01.

  Lohani, S., Kirby, B. T., Brodsky, M., Danaci, O., & Glasser, R. T. (2020). Machine learning assisted quantum state estimation. *Machine Learning: Science and Technology*, *1*(3), 035007.

  Zhong, L., Guo, C., & Wang, X. (2022). Quantum state tomography inspired by language modeling. *arXiv preprint arXiv:2212.04940*.

- The results depicted in Figure 5 appear somewhat counterintuitive. Why the reconstruction fidelity occasionally decreases as the number of samples increases?

---

> ### Author Response · Authors · 2023-11-21
>
> We thank the reviewer for his/her constructive comments. We apologize for overlooking the references and has added in the revised version. However, we note that circuit cutting is a promising tool to simulate large quantum circuits with smaller quantum computers (currently available ones in the market). In our paper, we are trying to mimick the results from the actual quantum computer and  try to reconstruct the states that would have obtained if it were to run on a large one.  The reference mentioned by the reviewers are for the full circuit. We have emphasized in the paper that due to the entanglement that can be generated in the quantum circuits, state reconstruction after circuit cutting is a non trival task. Hence, it is more demanding than without cutting a circuit.
>  The fidelity was calculated only for one training instance. We note that the fidelity will vary over 3% over the average value when the experiment is repeated for GHZ states. We have modified the text in the paper to note this.
> We also tested the circuit by inserting the random unitary gates distributed according to Haar measure. In this case, the fidelity increases with the increase in the sample size. This is because GHZ states has only two outcomes either all |00000..00> state or |111..111> state. However, with random unitary gates since the outcome can be random basis states like 00..0, 100..0, 010..0 etc. Hence, more samples helps to increase the fidelity. We have modified the Fig.5 to add the results with random unitary gates.

---

> > ### Comment · Reviewer_YKiP · 2023-11-23
> >
> > I thank the authors' efforts. I will maintain my rating.

---

### Official Review · Reviewer_DHPz · 2023-10-30

**Soundness:** 3 good
**Presentation:** 2 fair
**Contribution:** 2 fair
**Rating:** 3
**Confidence:** 4

**Summary:**

In this paper, the authors showcase efficient state reconstruction using a Restricted Boltzmann Machine (RBM) with polynomial resource scaling. This work can be viewed as an extension of RBM-based quantum state tomography, as previously published in Nature Physics. While the experimental results are truly inspiring, it is worth noting that for potential readers, primarily from EE and CS backgrounds at ICLR, the expressions and notations in this paper may pose challenges for comprehension.

**Strengths:**

The authors demonstrate efficient state reconstruction using a Restricted Boltzmann Machine (RBM) with polynomial resource scaling. The experiments illustrate that fragmented Greenberger–Horne–Zeilinger circuits, at the state-of-the-art scale of up to 18 qubits, can be reconstructed with near-perfect fidelity using only 100 sample measurements. This is a very encouraging result.

**Weaknesses:**

As previously mentioned, ICLR primarily caters to readers in the fields of EE and CS. While the results demonstrated in quantum machine learning are highly encouraging, the introduction of quantum circuits and fundamental mechanics can be challenging for those without a background in quantum physics. I recommend that the authors consider revising Section 3. Based on the next revision, I will contemplate raising my score.

**Questions:**

1. To facilitate a more straightforward understanding of the primary novelty, it is advisable to explicitly introduce the fundamentals of quantum mechanics in Section 3.1. Additionally, for the benefit of readers, a comparison between notations in linear algebra and those in quantum physics should be provided to enhance clarity.

2. The model of quantum state tomography should be introduced after the basic quantum mechanics.

3. I am curious whether the method employed by the authors ensures the physical structure of the density matrix, including properties like positive semidefiniteness and unit trace. If this is not the case, it is essential for the authors to highlight this aspect.

4. In addition to the method presented in this paper, the existence of low-dimensional structures, such as low-rankness [1] and matrix product operators [2], offers another avenue for decreasing the number of necessary repeated measurements. The reviewer recommends that the authors should consider introducing potential approaches for reducing the requirement for repeated measurements, both from hardware and theoretical perspectives.

[1] J. Haah, A. Harrow, Z. Ji, X. Wu, and N. Yu, “Sample-optimal tomography of quantum states,” IEEE Transactions on Information
Theory, vol. 63, no. 9, pp. 5628–5641, 2017.

[2] Zhen Qin, Casey Jameson, Zhexuan Gong, Michael B Wakin, and Zhihui Zhu.  “Stable tomography for structured quantum states,” arXiv preprint arXiv:2306.09432, 2023.

5. The tilde above $\Lambda$ is misaligned on Page 5.

---

> ### Author Response · Authors · 2023-11-21
>
> We thank the reviewer for his/her valuable comments and careful reading of our paper.
> We have modified section 3. Since our method is for circuit cutting, we have introduced the cutting procedure before the method for continuity.
> We note that our method is based on state vector and hence doesn't need the requirements of positive semidefiniteness and unit trace required for density matrix. We have included that in the conclusion where the advantages of our method is emphasized.
> We apologize for overlooking the references pointed out by the reviewer and has added it in the revised version. We also note that the above schemes are mainly for the density matrix represenation and cannot be directly applied to our method.
> We thank the reviewer for pointing out the typo in page 5 which we have fixed in the revised version.

---

### Official Review · Reviewer_GKSV · 2023-11-01

**Soundness:** 2 fair
**Presentation:** 3 good
**Contribution:** 2 fair
**Rating:** 5
**Confidence:** 3

**Summary:**

The authors apply restricted Boltzmann machines (RBMs) to the task of estimating quantum states, as a means of reducing the sample overhead required for quantum circuit cutting procedures. They use numerical experiments on preparation circuits for GHZ states of increasing sizes to show that a small number of samples is sufficient to obtain high-fidelity reconstruction of the target state.

**Strengths:**

* The application of RBMs for learning representing quantum fragments from limited samples seems like a good fit for the needs of quantum circuit cutting, and I would describe this paper as primarily applying existing methods in the context of a fortuitous new domain. This new application likely opens the door for other compressed representations here, including tensor networks and more general neural network states.

* The reported reduction in sample complexity is impressive, and the numerical results showing this are one of the main advertised benefits of the method (although see my concerns about these results below).

**Weaknesses:**

* The claimed reduction in number of required samples is impressive, but I'm not completely convinced by the numerical evidence supporting this claim. The main reason for concern here is the reported performance of the method vs. number of samples (Figure 5 and left panel of Figure 3), where the fidelity seems to degrade as the algorithm is given access to more measurement outcomes. The explanation for this strange behavior provided in the text ("As the sample size is increased, the fidelity drops due to the overfitting of the data") seems highly unlikely, as overfitting typically becomes _less_ of an issue as more training data is provided. In the similar work of [Perlin et al. (2021)] for example, the fidelity increases monotonically with sample count for every tomographic method tested, converging to a fidelity of 1 in the large sample limit for each case (see Fig. 5 of that work). In light of this, the fact that the authors' results doesn't show any meaningful improvement with greater access to samples is concerning.

* The experiments supporting this method are fairly weak, in that they only consider a single simple type of quantum circuit. Although the authors compare the findings of this experiment to those of [Perlin et al. (2021)] ("their reconstruction procedure requires 1 million samples whereas we attain the same fidelity with only 100 samples"), this comparison is misleading, as the prior work tested their method against families of random circuits, clearly a more difficult task. I would encourage the authors to at least generalize their single GHZ example to a family of random quantum states, for example by adding random SU(2) rotations in the middle of the GHZ preparation circuit.

* Having a direct comparison between the proposed method and past tomographic methods on the same quantum circuits would be helpful for assessing the performance of this method, as no comparison is offered to prior work that actually used this same family of quantum circuits.

**Questions:**

* Do you know why your experimental results show decreasing fidelity with increasing samples? The overfitting explanation given in the text is implausible, but there could also be something I'm missing here.

* Do your methods give similar results when tested on a wider variety of quantum circuits?

---

> ### Author Response · Authors · 2023-11-21
>
> We thank the reviewer for his/her constructive comments. The fidelity was calculated only for one training instance. We note that the fidelity will vary over 3% over the average value when the experiment is repeated for GHZ states. We have modified the text in the paper to note this.
> We also tested the circuit by inserting the random unitary gates distributed according to Haar measure. In this case, the fidelity increases with the increase in the sample size. This is because GHZ states has only two outcomes either all |00000..00> state or |111..111> state. However, with random unitary gates since the outcome can be random basis states like 00..0, 100..0, 010..0 etc. Hence, more samples helps to increase the fidelity. We have modified the Fig.3 to add the results with random unitary gates. We still see that the fidelity can be improved though one needs more samples compared to GHZ states.
> We note that comparison with Perlin et al. (2021) is not fair and hence we have removed the sentences from the paper where the comparison is made. However, we note that our method can accurately reconstruct even random states.